# Host-Parasite Interaction in *Sarcoptes scabiei* Infestation in Porcine Model with a Preliminary Note on Its Genetic Lineage from India

**DOI:** 10.3390/ani10122312

**Published:** 2020-12-07

**Authors:** Arun Kumar De, Sneha Sawhney, Samiran Mondal, Perumal Ponraj, Sanjay Kumar Ravi, Gopal Sarkar, Santanu Banik, Dhruba Malakar, Kangayan Muniswamy, Ashish Kumar, Arvind Kumar Tripathi, Asit Kumar Bera, Debasis Bhattacharya

**Affiliations:** 1Animal Science Division, ICAR-Central Island Agricultural Research Institute, Port Blair 744101, India; snehasawhney88@gmail.com (S.S.); perumalponraj@gmail.com (P.P.); skravivet@gmail.com (S.K.R.); swamy02_vet@yahoo.co.in (K.M.); akt9@rediffmail.com (A.K.T.); debasis63@rediffmail.com (D.B.); 2Department of Veterinary Pathology, West Bengal University of Animal and Fishery Sciences, Kolkata 700037, India; vetsamiran@gmail.com (S.M.); gsarkar999@gmail.com (G.S.); 3Department of Animal Genetics and Breeding, ICAR-National Research Centre on Pig, Guwahati 781131, India; sbanik2000@gmail.com; 4Animal Biotechnology Centre, National Dairy Research Institute, Karnal 132001, India; dhrubamalakar@gmail.com; 5CTARA, IIT Bombay, Mumbai 400076, India; ashish7@iitb.ac.in; 6Reservoir and Wetland Fisheries Division, ICAR-Central Inland Fishery Research Institute, Barrackpore, Kolkata 700120, India; asitmed2000@yahoo.com

**Keywords:** *Sarcoptes scabiei*, host-parasite interaction, molecular characterization, lipid profile, antioxidant

## Abstract

**Simple Summary:**

Scabies or mange caused by *Sarcoptess cabiei* is the latest addition of WHO’s list oftropical neglected diseases. It causes severe itching to the host. It has a wide host range including humans, farm animals, companion animals, and wild animals. It is anemerging/re-emerging disease with high prevalence in underdeveloped and developing countries. The disease has zoonotic importance and is of significant public health concern as cross-transmission or species jumping is very common. To date, fifteen *Sarcoptes* varieties have been reported as per host origin. Differential diagnosis at variety level is very crucial for epidemiological study and scratching future eradication program of the disease. As morphotaxonomy fails to differentiate varieties, use of molecular markers is crucial. Moreover, it is very important to understand the host-parasite interaction at the systemic level for a better understanding on the pathogenicity of the disease. Here, we report the genetic characterization of *S. scabiei* from India and host-parasite interaction in a porcine model.

**Abstract:**

The burrowing mite *Sarcoptes scabiei* causes scabies in humans or mange in animals. It infests a wide range of mammalian species including livestock, companion animals, wild animals, and humans. Differential diagnosis of *Sarcoptes* varieties is key for epidemiological studies and for formulation of an eradication program. Host-parasite interaction at the systemic level is very important to understand the pathogenicity of the mite. This communication deals with the preliminary report on the genetic characterization of *S. scabiei* from India. Moreover, the effect of *S. scabiei* infestation on host physiology with special emphasis on serum biochemical parameters, lipid profile, oxidant/antioxidant balance, stress parameters, and immune responses were evaluated in a porcine model. Cytochrome C oxidase 1 and voltage-sensitive sodium channel based phylogenetic study could distinguish human and animals isolates but could not distinguish host or geographical specific isolates belonging to animal origin. An absence of host-specific cluster among animal isolates argues against the hypothesis of delineating *S. scabiei* as per host origin. Elevated levels of markers of liver function such as albumin, AST, ALT, ALP, and LDH in infested animals indicated impaired liver function in infested animals. *S. scabiei* infestation induced atherogenic dyslipidemia indicated by elevated levels of total cholesterol, low-density lipoprotein cholesterol and triglycerides, and a decreased level of high-density lipoprotein cholesterol. Oxidative stress in infested animals was indicated by a high level of nitric oxide and serum MDA as oxidative stress markers and low antioxidant capacity. *S. scabiei* triggered stress response and elevated levels of serum cortisol and heat shock proteins were recorded in infested animals. *S. scabiei* infestation increased the serum concentration of immunoglobulins and was associated with up-regulation of IL-2, IFN-γ, IL-1β, and IL-4 indicating both Th1 and Th2 response. The results of the study will be helpful for a better understanding of host-parasite interaction at the systemic level in crusted scabies in pigs.

## 1. Introduction

Scabies in human, or mange in animals, caused by the burrowing mite *Sarcoptes scabiei*, is a neglected tropical infectious disease and is prevalent worldwide especially in developing and under-developed countries [1,2]. It infests more than 100 mammalian species including livestock, companion animals, wild animals, and humans [3,4,5]. Mange is considered as an emerging/re-emerging infectious disease and imparts huge economic loss to the livestock industry due to devastating morbidity and reduced production [3,5]. It is regarded as a significant public health concern in underdeveloped and developing countries as it is frequently associated with bacterial super-infection, which may sometimes be fatal especially in children if not treated in time [6]. Worldwide, around 300 million people are estimated to be infested with scabies every year [7] and its prevalence among children in Africa and indigenous communities of Northern Australia is astoundingly high ranging from 25 to 80% [8,9,10]. Scabies is broadly categorized into two forms, ordinary scabies with low mite burden (<20) or crusted scabies with high mite load and hyper-keratosis of the skin [11]. The mite *S. scabiei* causes inflammation of the skin which is associated with an exudate which forms crusts on the surface [12].

Controversy and conflicting opinion exist regarding speciation of *S. scabiei*; some believe that *S. scabiei* infesting different hosts are monospecific whereas others claim that they belong to different species or subspecies [13,14]. Currently, there is a broad consensus that the species *Sarcoptes* is divided into different varieties as per its host origin [13,15] and to date, more than 15 diverse varieties have been reported [2,16]. Of recent, this hypothesis has been challenged as cross-transmission or species jumping is very common in some varieties [17]. Therefore, differential diagnosis of *Sarcoptes* varieties is crucial for epidemiological studies [18] and formulating eradication strategies [15]. Morphotaxonomy has failed to differentiate between varieties as they share similar morphology [19]. Molecular diagnosis based on ribosomal DNA spacer region has been unsuccessful in identifying *S. scabiei* to the host level [20] and immunological diagnosis is also very challenging as different varieties produce immunologically identical proteins [21]. Recently, mitochondrial cytochrome oxidase 1 (COX1) based marker has shown promising results in varieties level confirmation of *S. scabiei* infestation. No information on genetic characterization or molecular confirmation in *S. scabiei* is available from India. The present study seems to be the first report on the genetic characterization of *S. scabiei* from India.

Sarcoptic mange is a very common ectoparasite disease in pigs [22,23]. It affects growth rate and reproductive performance in pigs and increases piglet mortality [24,25,26] leading to significant economic loss to the swine industry. A rare probability of zoonotic potential of *S. scabiei* var suis cannot be ignored [27]; it may be transmitted to affect a variety of different animal host species as well as pig handlers [28,29] causing severe itching. As pigs in semi-intensive systems in developing and under-developed countries are generally reared by tribes who keep a close association with the animals, there is a chance of transmission to humans, especially children and immunocompromised adults [4]. Moreover, in developed countries, pigs have been kept as pets which can transmit the disease to humans. Sarcoptic mange is a very common problem in the pig rearing regions of India, especially Northeast and Eastern parts of India [30]. In the Andaman and Nicobar Islands, Nicobari pigs are reared by Nicobarese tribal people and close association between tribal people and pigs is observed.

*S. scabiei* modulates host immunity, inflammatory, and complement reactions in order to be established to the host skin [31,32,33]. *S. scabiei* infestation triggers multiple reactions including allergic reactions, inflammation, innate immune reactions, and activation of immune components in the skin [34]. The salivary solutions of burrowing mites contain different bioactive compounds with potential to influence host physiological functions [35]. Mites are reported to influence the cytokine and chemokine secretion from keratinocytes and dermal fibroblasts [31,33] and disturb the balance between Th1 and Th2 immune responses [36]. In addition, *S. scabiei* has been reported to disturb the antioxidant defense system in mammalian hosts [37,38]. However, the studies were mostly in vitro using skin equivalents [31]. Little is known about the host parasite interaction in *S. scabiei* infestation at the systemic level. Therefore, the study aims at genetic characterization of *S. scabiei* isolated from Nicobari pigs and a better understanding on host parasite interactions with special emphasis on serum biochemical parameters, lipid profile, oxidant/antioxidant balance, stress parameters, and immune responses.

## 2. Materials and Methods

### 2.1. Ethics Approval

This research has been approved by the Institute Animal Ethics Committee (IAEC) of ICAR-Central Island Agricultural Research Institute (ICAR-CIARI), Port Blair, Andaman and Nicobar Islands, India on 12 January 2020 and the ethical approved project identification code is ‘ICAR-CIARI/AS/AICRP-Pig/IAEC/4960 dated 12 January 2020’. All the methods were performed in accordance with the relevant national guidelines and regulations.

### 2.2. Study Area and the Animals

The study was conducted on Nicobari pigs maintained at the institute pig farm of ICAR-CIARI (11.6060° N, 92.7058° E). Sarcoptic mange infestation in pigs was identified during a routine visit to the farm. Fifteen *S. scabiei*-infested animals (5–6 months old, 8 male and 7 female) participated in the present study. Ten healthy animals (5–6 months old, 5 male and 5 female) were treated as control. Absence of *S. scabiei* infestation in control animals was confirmed by dermatological and microscopic examination. All the animals were examined for presence of fleas, lice, ticks, or any other ectoparasite, and animals negative for these were only considered in the study. Coproscopic examination for light and heavy eggs [12] confirmed that the animals were free from gastrointestinal parasites and the animals were serologically negative for classical swine fever which is endemic to the Andaman and Nicobar Islands. Both control and infested animals were maintained in 12-h light-dark cycle on concrete floor in separate pens to avoid chance of transmission of sarcoptic mange and fed with a commercial diet containing 18% crude protein, 2% crude fat, 6% crude fiber, 4% acid insoluble ash and 10% moisture. The animals were offered water ad libitum.

### 2.3. Collection and Processing of Clinical Samples for Microscopic Analysis

Skin scrapings were collected from the infested Nicobari pigs using surgical blades. Mineral oil was applied on the surgical blades and scrapings were taken from the crusted area of skin till little blood appeared. Collected material was boiled in 10% sodium hydroxide (NaOH) (*w*/*v*) solution to dissolve the keratinized tissue and was centrifuged at 500× *g* for five minutes and the sediment was examined under 100× magnification in a light microscope (Trinocular Compound Microscope, Quasmo, Ambala, Haryana, India 180129/53268). Photographs of different stages of mite and eggs present were taken at 400× magnification.

### 2.4. DNA Extraction

DNA from skin scrapings was isolated by a commercial kit (DNeasy Blood and Tissue kit, Cat. No. 69504, Qiagen, Hilden, Germany). Briefly, 25 mg of skin scrapping was ground in lysis buffer using a sterile pestle and motor. For cell lysis, proteinase K was added to the ground scrapings and it was incubated at 56 °C in a water bath overnight. The completely lysed samples were used for genomic DNA extraction as per protocol recommended by the manufacturer. The DNA samples were kept at −80 °C until further use.

### 2.5. Amplification and Sequencing of Cytochrome C Oxidase Subunit 1 (COX1) and Voltage Sensitive Sodium Channel (VSSC) Gene Segments

For molecular identification and characterization of the mite, two gene segments namely COX1 and VSCC were chosen [34]. COX1 and VSCC gene fragments were amplified using primers described earlier by Erster et al. [34]. The PCR was performed in Thermocycler (Eppendorf, Hamburg, Germany) with the cycling conditions mentioned previously [34]. The amplicons were purified using a commercial kit (MinElute PCR Purification Kit, Cat. No. 28004, Qiagen, Hilden, Germany) as per manufacturer’s protocol and sequence information was generated by dideoxy fingerprinting.

### 2.6. Sequence Analysis

Representative COX1 and VSSC sequences of *S. scabiei* from different hosts distributed in different geographical regions were retrieved from GenBank (www.ncbi.nlm.nih.gov) and a summary of the information has been depicted in Appendix A. Alignment of the sequences was done by ClustalW [39] in MEGAX [40]. Phylogenetic tree was constructed in MEGAX using Maximum Likelihood method and Tamura–Nei model [41]. The reliability of the tree was judged by 1000 bootstrap replications. For phylogenetic tree construction, we trimmed extra nucleotides from our sequences and GenBank retrieved sequences to make a homogeneous length of 193 bp for COX1 and 436 bp for VSSC. Evolutionary relationship among different sequences was deduced by median-joining networks constructed in Network v 10 with default settings [42].

### 2.7. Estimation of Biochemical Parameters

Commercially available kits (TBL, Solan, India) were used for estimation of serum biochemical parameters such as total protein, albumin, globulin, glucose, aspartate aminotransferase (AST), alanine aminotransferase (ALT), alkaline phosphatase (ALP), lactate dehydrogenase (LDH) and creatine kinase (CK). The parameters were assessed in an automated clinical chemistry analyzer (Transasia Biomedical Limited, Mumbai, India). The reference values of the parameters are presented in Appendix A.

### 2.8. Estimation of Oxidative Stress Parameters

To investigate the oxidant/antioxidant balance in serum following *S. scabiei* infestation, the markers of oxidative stress such as serum level of total nitric oxide (NO), total serum antioxidant activity, malonyldialdehyde (MDA) concentration, as well as the levels of antioxidant enzymes; superoxide dismutase (SOD), glutathione-S-transferase (GSH), catalase were determined and compared to control animals.

NO level was evaluated by a commercial kit (EZAssay^TM^ Nitric Oxide Estimation kit, HiMedia Laboratories Pvt. Ltd., Nashik, India) according to the manufacturer’s instructions. The principle of the assay is the reduction of NO_3_ which was then converted to a blue-colored azo compound. The absorbance was recorded at 630 nm.

Antioxidant activity in serum was estimated by a colorimetric kit (EZAssay^TM^ Antioxidant Activity Estimation kit, HiMedia Laboratories Pvt. Ltd., Nashik, India) following the protocol recommended by the manufacturer.

Level of lipid peroxidation was measured by measurement of malonyldialdehyde (MDA). Levels of MDA in serum were measured by a commercial colorimetric based assay kit (EZAssay^TM^ TBARS estimation kit for lipid peroxidation, HiMedia Laboratories Pvt. Ltd., Nashik, India).

Serum levels of catalase, superoxide dismutase, glutathione-S-transferase were measured by ELISA based methodology using catalase assay kit, superoxide dismutase assay kit, and glutathione assay kit (Cayman chemicals, Ann Arbor, MI, USA) as per manufacturer’s protocol.

### 2.9. Estimation of HSPs in Serum

Serum levels of four heat shock proteins (HSP20, HSP40, HSP70, and HSP90) in infested and control animals were measured by ELISA-based methodology on biotin double antibody sandwich technology using commercial kits (Porcine HSP20, HSP40, SHP70 and HSP90 ELISA kit, Arsh Biotech Pvt. Ltd., Life Technologies, Delhi, India). Briefly, serum samples were added to the wells pre-coated with the respective HSP monoclonal antibodies. This was followed by the addition of respective anti-HSP antibodies labeled with biotin-streptavidin-HRP complex. Washing was done to remove the unbound proteins. This was followed by the addition of substrates and measurement of absorbance at 450 nm using a microplate reader (SpectraMax Plus, Molecular Devices, San Jose, CA, USA).

### 2.10. Measurement of Cortisol in Serum

Cortisol level in serum was measured by ELISA, based on biotin double antibody sandwich technology using a commercial kit (Porcine Cortisol ELISA kit, Arsh Biotech Pvt. Ltd., Life Technologies, Delhi, India) as per the protocol of the manufacturer. Briefly, serum samples were added to cortisol monoclonal antibodies pre-coated wells. This was followed by the addition of anti-cortisol antibodies labeled with biotin-streptavidin-HRP complex. Washing was done to remove the unbound proteins. This was followed by the addition of substrates and measurement of absorbance at 450 nm using a microplate reader (SpectraMax Plus, Molecular Devices, San Jose, CA, USA). The reference value of cortisol is presented in Appendix A.

### 2.11. Lipid Profile Analysis

Serum levels of total cholesterol (TC), high-density lipoprotein cholesterol (HDLc), low-density lipoprotein cholesterol (LDLc), and triglycerides (TG) were evaluated. Levels of TC, HDLc, and TG were determined by enzymatic methods using commercially available kits (Jeev Diagnostics Pvt. Ltd., Chennai, India; Pathozyme Diagnostics, Kholapur, India; and Spinreact, S.A., Spain respectively). The concentration of LDL was deduced by the following formula: LDL = TC − HDL − (TG ÷ 5) [43]. Moreover, we calculated cardiac risk factor (CRF) and atherogenic index (AI) using the following formulas: CRF =TC/HDL [44] and AI = (TC − HDL)/HDL [45]. The reference values of the parameters are presented in Appendix A.

### 2.12. Estimation of Serum Immune Parameters

The concentrations of total IgG, IgA, and IgM in serum were determined using quantitative ELISA kits (Arsh Biotech Pvt. Ltd., Delhi, India) according to the manufacturer’s instructions.

Serum levels of cytokines (IL-2, IL-4, IL6, IL-8, IL-12, IL-1β, IFN-γ) in the serum of *S. scabiei*-infested and control animals were measured by sandwich ELISA methodology using commercial kits from Life Technologies, New Delhi, India.

### 2.13. Estimation of Apoptotic Markers

Serum concentrations of three apoptotic markers (Caspase-3, Caspase-7, and Casepase-8) were estimated by Nori^®^ porcine ELISA kits (Life Technologies Pvt. Ltd., Delhi, India) according to the manufacturer′s manual.

### 2.14. Histopathology

Skin scrapings from infested animals were collected and fixed in formalin solution (10%) at room temperature and a routine process was adopted for histopathology. In brief, samples were passed through ascending grades of alcohol (70–100%) for dehydration. The dehydrated samples were then cleared in benzene, transferred to melted paraffin (60 °C) for impregnation, and paraffin blocks were finally prepared in metal molds. Sections were prepared by using microtome (Leica, Wetzlar, Germany; Catalogue No. RM2235) at 5μM Slides were prepared as per standard protocol and were stained with hematoxylin and eosin.

### 2.15. Statistical Analysis

Data were presented as mean ± standard deviation and were normally distributed. Differences among control and infested groups were calculated by *t*-test using GraphPad Prism software (San Diego, CA, USA) (http://www.graphpad.com). The mean values with a significance of *p* < 0.05 were considered to be statistically significant.

## 3. Results

### 3.1. Clinical Signs of the Infested Animals

Clinical examination showed intense pruritus associated with hyperkeratosis and crusts in the skin. Skin of the animals was thickened and wrinkled in appearance which was a characteristic feature of crusted scabies/Norwegian scabies. Alopecia in the affected area was observed (Figure 1a).

### 3.2. Microscopic Examination

On the basis of clinical symptoms, skin scrapings from infested animals were examined under microscope for the presence of mites. Microscopic examination confirmed the presence of eggs, eggs with six-legged larvae, and nymphal stage of *S. scabiei* (Figure 1b–e). The nymphal stages had short legs. The third and fourth pair of legs never projected beyond the body. Morphologically the mites were indistinguishable from *Sarcoptes* [12].

### 3.3. Molecular Identification and Characterization of S. scabiei

For molecular confirmation, one mitochondrial gene (COX1) and one nuclear gene (VSSC) were amplified using *Sarcoptes* specific primers and amplicons of both the genes were detected in infested pigs whereas in control pig, no amplicons were observed (Figure 1f). Specificity of the PCR was verified by using pig lice (*Haematopinus suis*) DNA as a negative control for both the genes (Figure 1f).

For differential diagnosis and molecular characterization, we generated partial sequence information of COX1 (2 samples) and VSSC (one sample) and the sequences were deposited in GenBank and obtained accession number MN986997-MN986998 for COX1 and MN986999 for VSSC. The sequences of COX1 and VSSC were 686 bp and 498 bp in length respectively. For phylogenetic analysis, we retrieved representative *S. scabiei* COX1 and VSSC sequences from different hosts distributed in different geographical regions. COX1 based phylogenetic tree (Figure 2a) indicated three distinct clades; two for human isolates (Clade A and B) and one for animal isolates (Clade C). In Clade A, two clearly defined clusters were detected; one includes human isolates from Hong Kong, South Korea, China and the other includes human isolates from France and Australia. Clade B contains human isolates from Panama. On the contrary, no host or region-specific clusters were observed in animal clade (Clade C). From the phylogenetic tree, it was evident that the Andaman isolates were phylogenetically close to pig isolate of Israel. A median-joining network was constructed to understand the evolutionary relationship of *S. scabiei* isolates. Two human-specific clades and one animal-specific clade were detected (Figure 2b).

Phylogenetic tree based on VSSC sequences showed three host-specific distinct clusters, one for human isolate (Cluster A) and the other two for animal isolates (Figure 2c). Within animal isolates, pig and erinaceus isolates formed one cluster (Cluster B) whereas oryctolagus, golden jackal, vulpes, and canis isolates formed another cluster (Cluster C). Pig isolates of Andaman belonged to Cluster B with pig isolates from Israel, Australia, and erinaceus isolate from Israel. The VSSC based network tree is presented in Figure 2d and three clearly defined clusters were depicted.

### 3.4. Serum Biochemical Parameters in Crusted Scabies

A significant decrease in plasma, total protein, albumin, and glucose, and significant increase in globulin, ASL, ALT, CK, LDL, and ALP in infested group as compared to those of control group was observed (Figure 3). Moreover, the ration of albumin and globulin (A:G) in infested group (0.6359 ± 0.048) was decreased than that of control group (1.121 ± 0.066).

### 3.5. Crusted Scabies Leads to Dyslipidemia in Pigs

The results of the lipid profile analysis are presented in Figure 4. In the infested group, an atherogenic dyslipidemic profile indicated by elevated levels of total cholesterol, low-density lipoprotein cholesterol (LDL-c) and triglycerides (TG), and a decreased level of high-density lipoprotein cholesterol (HDLc) as compared to control group was observed. This was further supported by higher CRF and AI values in infested animals than those of control animals.

### 3.6. Crusted Scabies Induces Oxidative Stress

To investigate the effect of *S. scabiei* infestation on oxidant/antioxidant balance in serum, the markers of oxidative stress such as total antioxidant activity (T-AOC), total nitric oxide (TNO) concentration, levels of lipid peroxides, and malonyldialdehyde (MDA) as well as the levels of antioxidant enzymes; superoxide dismutase (SOD), glutathione-S-transferase (GSH) and catalase were determined and compared to control animals (Figure 5). The infested group showed significantly higher serum total nitric oxide (TNO) concentration than the control group. Total antioxidant activity (T-AOC) in the infested group was significantly lower than that of control group. Activity of catalase, SOD, and concentration of GSH in infested animals decreased significantly as compared to those of control animals. On the other hand, MDA concentration in the infested group was found significantly higher than its corresponding value in the control group (Figure 5).

### 3.7. Crusted Scabies Is Associated with Stress

To investigate whether *S. scabiei* infestation leads to stress in host animals, level of stress biomarkers such as serum cortisol concentration and serum levels of four heat shock proteins (HSP20, HSP40, HSP70, and HSP90) in control and infested animals were analyzed. Serum cortisol concentration in infested group was found significantly higher than that of control group (Figure 6).

HSP20, HSP70, and HSP90 were up-regulated in infested animals compared with the control animals. On the other hand, no significant change in HSP40 level between control and infested animals was detected (Figure 7).

### 3.8. Crusted Scabies Up-Regulates Serum Levels of Apoptotic Markers

Serum concentrations of all three apoptotic markers (Caspase 3, 7, and 8) in infested and control animals were recorded. Up-regulation of all three apoptotic markers was detected in infested group as compared to control group (Figure 8).

### 3.9. Crusted Scabies Alters Immune Response

Effect of *S. scabiei* infestation on the concentration of immunoglobulins and the level of cytokines were evaluated. Significantly higher concentrations of serum IgA, IgG, and IgM were detected in the infested group as compared to those of the control group (Figure 9a–c). Up-regulation of pro-inflammatory cytokines (IL-2, IL-6, IL-12, IL-1β, and IFN-γ) and anti-inflammatory cytokine IL-4 were detected in the infested group than the control group (Figure 9d–j). On the other hand, no significant difference in IL-8 concentration between the two groups was detected (Figure 9g).

### 3.10. Histopathology

Parakeratotic hyperkeratosis characterized by thickening of the stratum corneum and the presence of nuclei was observed. Mite with its exoskeleton and remnants were detected in the tunnel of stratum corneum of the epidermis and appeared as a cleft in the upper epidermis. Epidermis exhibited acanthosis and spongiosis associated with dense eosinophilic dermal infiltrate. Superficial perivascular or diffuse infiltrate of lymphocytes and histiocytes, accompanied by neutrophils and eosinophils were evident in the dermis. There was psoriasiform hyperplasia characterized by epidermal projections into the dermis for interdigitaion with dermal papillae. Excessive production of squames suggested crust formation (Figure 10).

## 4. Discussion

Scabies is considered a disease of resource-poor communities of underdeveloped and developing countries and a neglected tropical disease [46]. Wide host range and presence of different varieties necessitate differential identification for epidemiological study and for designing eradication strategy. *S. scabiei* infestation in multiple hosts including humans, companion animals, farm animals, lab animals, and wild animals throughout the globe points toward pathogen dispersal and spillover [47,48]. Bidirectional interactions among human, animals, pathogen, and environment might be the underlying reason behind its multi-host adaptation [49]. As morphological and serological tests are not much successful at variety level identification, molecular markers have emerged as alternative differential diagnostic tools. Mitochondrial cytochrome C oxidase subunit I gene (COX1) is a well-established universal marker for species identification due to its high variability [50,51] and is being used extensively for DNA bar-coding of medically important parasites [52]. Erster et al. [34] used nuclear marker VSSC for genetic characterization of *S. scabiei* from multiple hosts in Israel. In the present study, we used COX1 and VSSC markers for molecular characterization of the *S. scabiei* isolated from pig host. COX1 based phylogenetic tree (Figure 2a) indicated that isolates under animal clade (Clade C) were phylogenetically closer to Clade B (human isolates from Panama) than Clade A (human isolates from Hong Kong, South Korea, China, France, and Australia). In general, human isolates of *S. scabiei* are heterogeneous in nature, and proximity of Clade B and Clade C may be due to low gene flow between mites of those groups [53]. COX1 and VSSC based phylogenetic study could not distinguish host or geographical specific isolates belonging to animal origin. Absence of host-specific cluster among animal isolates argues against the hypothesis of delineating *S. scabiei* as per host origin and provides support to the hypothesis that *S. scabiei* from different hosts and different geographical location arose from a single speciation event [16,54,55]. Evolution of different varieties in different hosts might be due to mitochondrial capture or selective sweep phenomenon which is considered as the driving force behind uniparental inheritance [56]. This hypothesis has been proposed as the adaptation of *Sarcoptes* in different marsupials in Australia [48]. Moreover, parasites with low independent dispersal capacities face strong population bottlenecks; under such situations they lower their genetic diversity to maintain their adaptive potential [57,58]. This might explain the low genetic diversity in *S. scabiei* among animal isolates.

*S. scabiei* is a contagious disease with a broad host range of more than 100 species and its effects across species are generally conserved [59]. Therefore, it is reasonable to assume that understanding the changes in one host may have significance in other hosts [60]. In recent years, pigs have gained popularity as a biomedical model in translational research due to its anatomical and physiological similarity with humans [61,62]. Close resemblance between human and pig in clinical manifestation and disease progression of *S. scabiei* [63,64] underscore the importance of porcine model in understanding host-parasite interactions of scabies/mange. The present study showed marked changes in serum biochemical parameters, lipid profiles, stress parameters, oxidant/antioxidant balance, and immune parameters following crusted scabies in a porcine model.

Increased ALT, AST, ALP, LDH, CK, and decreased albumin and glucose levels in infested animals indicate organ damage, especially compromised liver function [65]. Oxidative stress has been associated with cell necrosis and apoptosis through activation of several cell signaling pathways such as JNK, mitogen-activated protein kinase (MAPK) leading to cellular and organ damage [66,67]. It is a well-established fact that oxidative stress causes liver injury [68] as the liver is a major target organ of ROS [69]. Moreover, free radicals, especially ROS catalyze the oxidation of LDL to generate oxidized LDL and oxidized LDL also promotes cellular apoptosis in different organs [67,70]. In the present study, crusted scabies induced oxidative stress in pigs which might have detrimental effects on organs including the liver. Oxidative stress-induced organ damage might explain elevated levels of AST, ALT, and alkaline phosphatase. Alteration in the biochemical parameters was also reported in sarcoptic mange affected goats and dogs [37,38,71]. As the liver is the major organ for albumin synthesis and glucose generation through gluconeogenesis and glucogenolysis [72], impaired liver function may attribute low levels of glucose and albumin in infested animals. In addition, high ROS has been reported to impair insulin signaling [73,74] causing an increase in insulin secretion; which might be another reason for low glucose concentration in infested animals. On the contrary, up-regulation in globulin production in infested animals might be due to the production of antibodies against *S. scabiei* antigens [38]. Low glucose and albumin and high globulin were also reported in Sarcoptes-infested goats [37] and dogs [38].

Abnormal blood lipid metabolites including elevated TC, LDLc, and TG were detected in infested animals than control animals. Imbalance in lipid metabolites is considered to be a predisposing cause of atherosclerosis and cardiovascular disorders [75]. In the present study, atherogenic dyslipidemia associated with high CRF and AI values in infested animals is an indicator of their susceptibility towards cardiovascular diseases. Imbalance in lipid metabolism is associated with many dermatological disorders especially chronic inflammatory skin diseases such as psoriasis, litchen planus, granuloma annulare, and histiocytosis [76]. Chronic inflammation in animals with crusted scabies may explain the abnormal lipid profiles. Moreover, oxidative stress which was also observed in the present study induced production of TNF-Alpha by Kupffer cells, which might induce inflammation [68]. Increased levels of pro-inflammatory cytokines following infestation might be the underlying cause behind dyslipidemia [76]. Increased serum levels of pro-inflammatory cytokines such as IL-2, IL-6, IL-12, IL-1β, and IFN-γ in infested animals were detected in the present study. Pro-inflammatory cytokines such as IL-6, IFN-γ inhibit the activity of lipoprotein lipase (LPL) [77] which hampers the clearance of VLDL and LDL cholesterol [78,79] leading to their increased concentration in blood. Moreover, crusted scabies induced oxidative stress in animals, and under oxidative stress, ROS induced activation of atherogenic genes via NF-kappa B [67] may further explain the atherogenic dyslipidemia in infested animals. In addition, alopecia due to infestation results in disruption of pelt-environment interface which in turn hampers thermoregulation as excessive heat loss to the environment occurs through the skin. This creates a negative energy balance to the host [60]. Mobilization of fat from fat stores to meet the energy deficiency may induce imbalance in lipid profile. An imbalance in fatty acid composition in adipose tissue with up-regulated omega-6 acids and down-regulated omega-3 acids in *S. scabiei* infested wombat host was reported [60]. The results of the study indicate that crusted scabies may shift lipid profile towards atherogenic dyslipidemia with high susceptibility towards cardiovascular disorder.

We determined the serum concentration of cortisol which is considered a biomarker of stress. The study found a significant increase in serum cortisol concentration in infested animals than control animals which indicated that infestation triggered stress response. Hypothalamic-pituitary-adrenal (HPA) axis, an important hormonal regulator of stress, is thought to be involved in the production of cortisol hormone [80]. In stressful conditions, activation of HPA axis followed by elevated plasma cortisol level is an adaptive response of the host [81]. HSPs are constitutively produced in normal physiological conditions whereas stress of any kind including environmental stress, viral infection, glucose deprivation, exposure of toxins, and oxidative stress induce over-expression of HSPs or cell-stress proteins [82]. HSPs are a classic example of proteins with ‘moonlighting functions’ [83] as they are involved in a number of physiological functions. Besides acting as chaperone and protein folding, involvement of HSPs in immune function such as antigen processing and presentation, expression of innate receptors and stimulation of innate immunity is well documented [84]. Moreover, HSPs, especially HSP70 and 27 play vital roles in stress-induced DNA repair by modulating DNA-repair enzymes, thus maintaining genome integrity [84]. Stress up-regulates the intracellular production of HSPs and they are released into the circulation following a non-classical secretion pathway [85]. In the present study, crusted scabies induced cellular and oxidative stress which in turn might have stimulated over-expression of HSPs. Thus, HSP orchestrated activation of heat shock response is an adaptation mechanism under stress [86]. Oxidative stress shifts the intracellular environment towards the oxidative state which in turn disturbs the protein native conformation and proteins lose their folded structure [86]. HSPs act as sensors to cellular redox changes both in prokaryotes and in eukaryotes; then give signals for activation of the heat stress response [86]. HSP70 and HSP90 act as the primary sensors of protein misfolding. Moreover, HSP27 has been reported to act as an antioxidant by preventing oxidation of glutathione under oxidative stress [87]. HSP70 and HSP90 act as anti-apoptotic proteins; they exert their role by binding to apoptosis protease activity factor 1 (Apaf-1) blocking downstream cascades of apoptosis pathway [88,89]. In the present study, the infested animals under oxidative stress might have induced cellular apoptosis. This is supported by the up-regulation of serum levels of apoptotic markers in infested animals. The up-regulation of HSPs in infested animals might be an adaptation mechanism to prevent cellular apoptosis.

Oxidative stress has been reported to be involved in the pathogenesis of several diseases including parasitic infections [90]. Cellular metabolism produces reactive oxygen species (ROS) and reactive nitrogen species (RNS) [91]. At a low to moderate level, they are involved in several physiological functions including cell signaling as well as immune functions [92]. On the contrary, at a high level, they cause damage to molecules including lipids, proteins, lipoproteins, and nucleic acids starting a chain reaction of free radical formation leading to a condition known as oxidative stress [93,94]. Therefore, tight regulation in production of ROS and RNS and delicate balance between beneficial and detrimental effects is vital for cellular homeostasis. Under oxidative stress, the antioxidant defense system elicits several antioxidant enzymes to fight against free radicals and maintain homeostasis. Superoxide dismutase, catalase, and glutathione peroxidase act as first-line defense antioxidants to suppress or prevent the formation of reactive species in cells [95]. SOD catalyzed the conversion of O_2_^−^ into H_2_O_2_ which is further reduced to H_2_O and O_2_ by either catalase or glutathione peroxidase [96]. In the present study, it was observed that crusted scabies caused an increase in the production of serum total NO in pigs. Though we did not measure ROS, oxidative stress induction in infested animals was evident from the reduction of total antioxidant capacity, activity of catalase, SOD, and concentration of GSH as well as an increase in MDA concentration as compared to control animals. The down-regulation of T-AOC and antioxidant enzymes indicate that they have been over-consumed to tackle oxidative stress [97]. The results of our study are in agreement with the report of Beighet al. [98], in which decreased catalase and GSH in dogs suffering from dermatophytosis was reported. A decrease in antioxidant enzyme activities was also reported in buffaloes infested by sarcoptic mange [99] and in sheep, which was suffering from psoroptic mange [100]. Increased MDA concentration in infested animals indicated oxidative injury. Cell membranes with a rich store of polyunsaturated fatty acids are excellent targets for free radical attacks [101] and oxidation of lipid molecules induce production of MDA [102]. So, MDA, the end product of lipid peroxidation, is considered as a hallmark of oxidative damage [103]. In the present study, increased MDA concentration in infested animals might be associated with cellular damage. An increase in skin MDA in *S. scabiei* infested dogs were reported by Nwufoh et al. [104].

*S. scabiei* infestation is associated with modulation of various parameters of host immune responses; humoral, adaptive, and inflammatory immune responses [31,33]. IgA, IgG, and IgM are the major immunoglobulins associated with humoral immunity. IgM is produced at the early stage of antibody response against a foreign antigen. IgG, being the most abundant antibody in blood and extracellular fluid, is involved in systemic immune response. IgA is generally associated with mucosal immunity and primarily found in secretions [105]. In the present study, up-regulation of total IgA, IgG, and IgM in infested pigs than the control group was observed. Elevated levels of immuniglobulins in infested animals indicated that mite had elicited antibody-mediated immune response against mite antigens although it was not possible to know whether or not the antibody response was *S. scabiei* specific. Up-regulation of total IgG might be a consequence of secondary bacterial infection which is very common in scabies infestation. Increased IgG in *S. scabiei* infested rabbits, dogs [21,106], and human [107] was reported. Elevated levels of circulatory IgA were reported in human crusted scabies [107].

We further assessed the systemic inflammatory response of the animals by determining serum levels of pro-inflammatory cytokines including IL-1β, IL-2, IL-6, IL-12, and IFN-γ and anti-inflammatory cytokine IL-4. Cytokines play key roles in the host immune and inflammatory responses as well as in the maintenance of tissue integrity [108]. The present study recorded dysregulation of cytokine balance which might be due to modulation of immunity by *S. scabiei* released antigens. Th1 and Th2 are the two major subpopulations of T helper cells and parasitic infections are associated with either Th1 or Th2 polarized immune response [109]. IL-2, TNF-β, and IFN-γ are Th1 cells specific cytokines whereas Th2 cells release IL-4, IL-5, IL-10, and IL-13 as signature cytokines [110]. Naïve helper T cells (Th0) can be differentiated to either Th1 or Th2 based on distinct activation pathways; cytokines such as IFN-γ andIL-12 activates Th0 to Th1 cells, otherwise, IL-10 and IL-4 are crucial stimulatory cytokines for Th2 cell polarization [111]. Intricate balance among different cytokines is crucial for homeostasis of mammalian species and imbalance may pose threat to health [112]. In the present study, *S. scabiei* infestation was associated with up-regulation of IL-2, IFN-γ, IL-1β, and IL-4 indicating both Th1 and Th2 response. The most probable cause of the mixed Th1/Th2 response might be dysregulation of cytokine balance due to modulation of immunity by *S. scabiei* released antigens or changes of cytokine levels after a chronic infection. Th1 immune response is generally associated with protection in most infectious diseases whereas Th2 immune response elicits high titers of antibody production and cell-mediated inflammation [113]. Moreover, allergic inflammatory diseases are dominated by Th2 immune response [6]. Induction of mixed Th1/Th2 immune response in the current study indicates that the immune response to mange infestation is complex and local immune response analysis is necessary for a better understanding of the disease. Th2 response had been described as a predominant immune response in crusted scabies and other allergic inflammatory disorders [6]. Bioactive compounds in arthropod saliva trigger different immune response to host. Low molecular weight salivary components cannot act as antigen but can bind to skin proteins as haptens and stimulate Th1 response. Some salivary antigens may cause basophil hypersensitivity (Th1 response) by binding to epidermal langerhans cells. Salivary antigens also may trigger Th2 response in association with IgE production and type I hypersensitivity [107].

The histopathological findings in the study mimic the classical narrative of sarcoptic mange in mammalian species [3]. Antigenic materials including excretion and secretions of mites trigger hypersensitivity reactions in the skin [114] which might be the reason behind eosinophilia observed in the infested animals. Our observations are consistent with those in vulpes [115], racoon dog [116], wombats [117] and Iberian ibex [118].

## 5. Conclusions

This communication is the first report on genetic characterization of *S. scabiei* from India. Mitochondrial (COX1) or nuclear (VSSC) based markers could not distinguish *S. scabiei* at a variety level especially for animal isolates which suggests that delineating varieties based on host origin is not warranted. The study contributes to the rich pool of knowledge on the consequences of crusted scabies on host physiology. We could establish a new connection showing that mange infestation results in an atherogenic dyslipidemia in the host.

## Figures and Tables

**Figure 1 animals-10-02312-f001:**
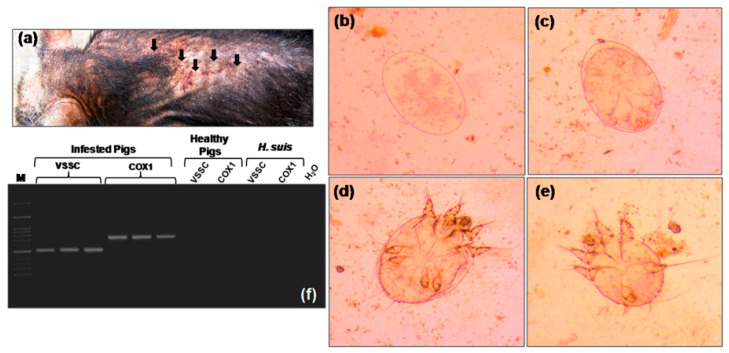
*Sarcoptes scabiei* infestation in Nicobari pig. (**a**) An infested animal with thickened and wrinkled skin and hair loss (black arrows), (**b**) mite egg (400× magnification), (**c**) mite egg containing larvae (400× magnification), (**d**) mite nymph (400× magnification), (**e**) larval stage of mite (400× magnification), (**f**) amplification of cytochrome C oxidase subunit 1 (COX1) and voltage sensitive sodium channel (VSSC) gene segments. Infested animals were positive for both of the gene fragments whereas control animals were negative. Primer specificity was verified using the lice (*Haematopinus suis*) DNA as negative control.

**Figure 2 animals-10-02312-f002:**
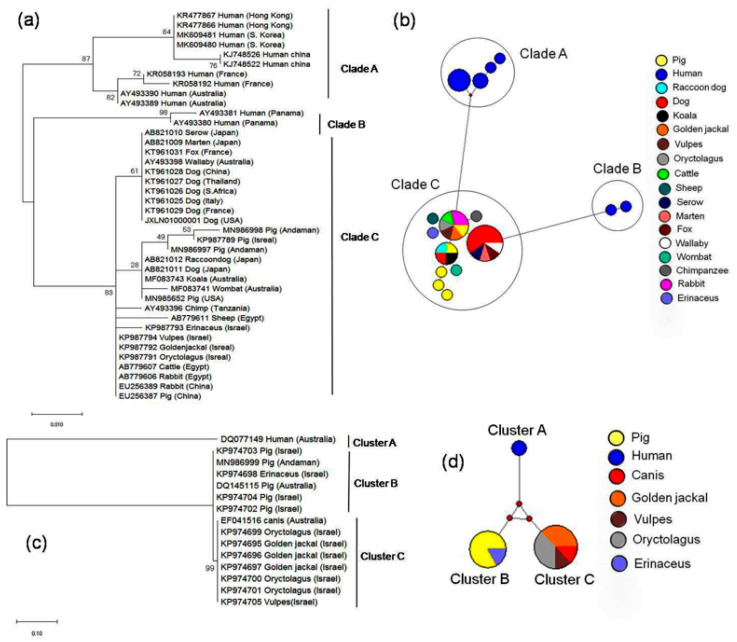
Evolutionary relationship of different isolates of *S. scabiei*. (**a**) COX1 based phylogenetic tree, (**b**) COX1 based network profile, (**c**) VSSC based phylogenetic tree, (**d**) VSSC-based network profile. Phylogenetic tree was constructed based on maximum likelihood method using Tamura–Nei model [41] implemented in MEGAX following 1000 bootstrap replications. Network was drawn in Network 10 with default settings [42].

**Figure 3 animals-10-02312-f003:**
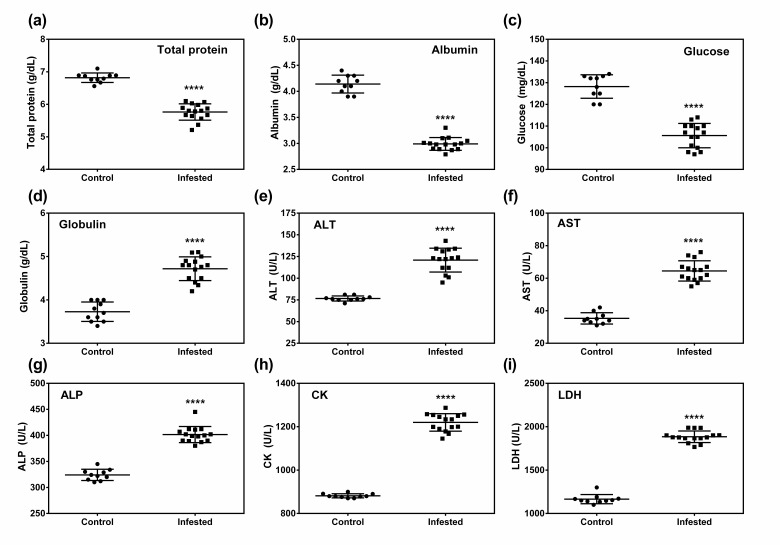
Effect of crusted scabieson serum biochemical parameters in a porcine model. (**a**) Serum total protein, (**b**) serum albumin, (**c**) serum glucose, (**d**) serum globulin, (**e**) alanine aminotransferase (ALT), (**f**) aspartate aminotransferase (AST), (**g**) alkaline phosphatase (ALP), (**h**) creatine kinase (CK), (**i**) lactate dehydrogenase (LDH). Data in scatter plot are shown as mean ± SD. *t*-test was performed to find out significant difference between the two groups. **** denotes *p* ≤ 0.0001.

**Figure 4 animals-10-02312-f004:**
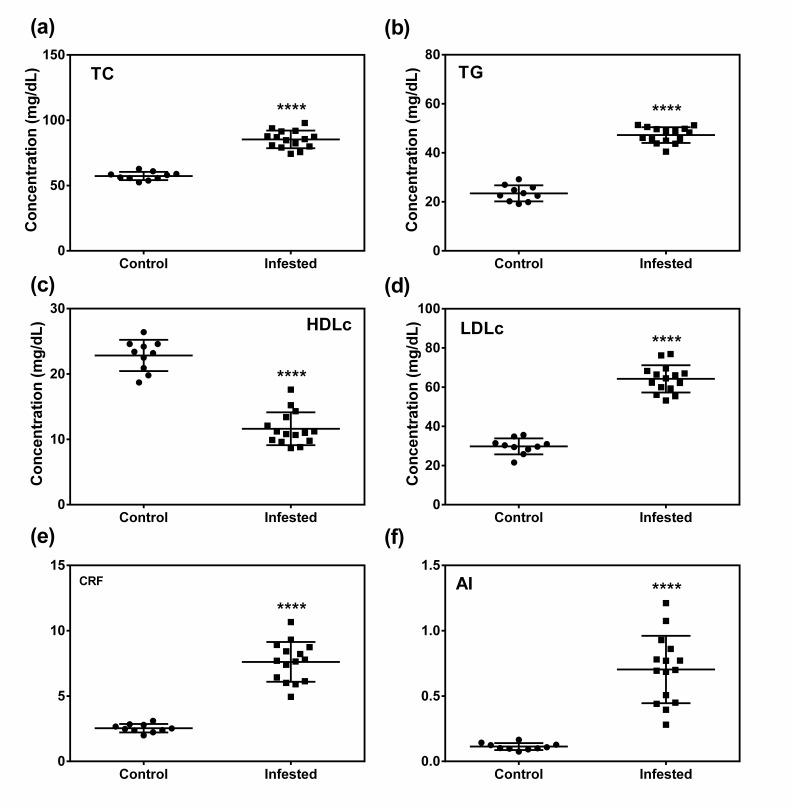
Effect of crusted scabies on serum lipid profile in a porcine model. (**a**) Total cholesterol (TC) concentration, (**b**) triglycerides (TG) concentration, (**c**) high-density lipoprotein cholesterol (HDLc) concentration, (**d**) low-density lipoprotein cholesterol (LDLc) concentration, (**e**) cardiac risk factor (CRF), (**f**) atherogenic index (AI). Data in scatter plot are shown as mean ± SD. *t*-test was performed to find out significant difference between the two groups. **** denotes *p* ≤ 0.0001.

**Figure 5 animals-10-02312-f005:**
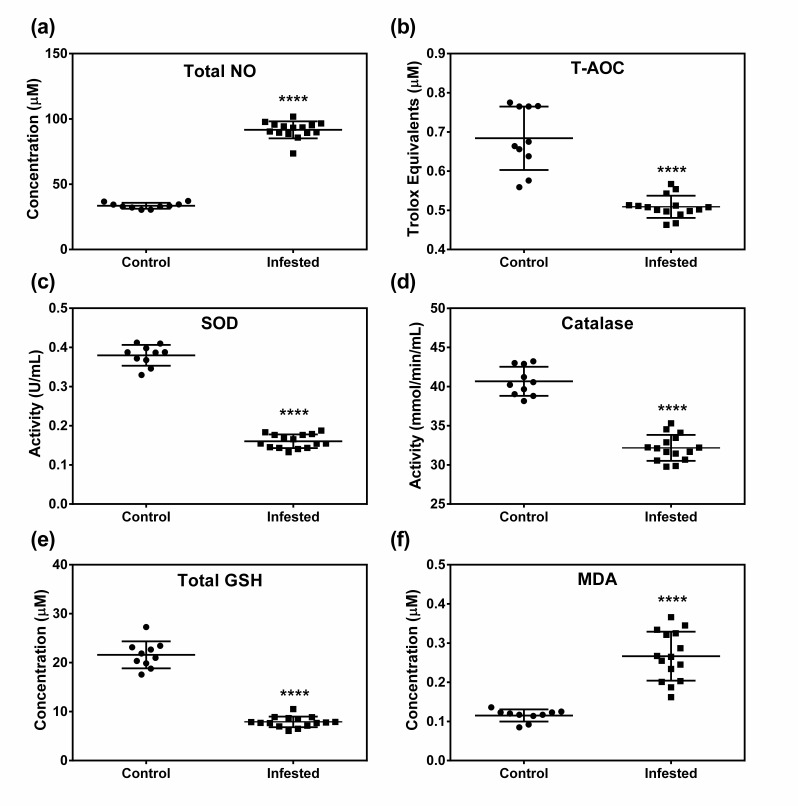
Effect of crusted scabies on antioxidant profiles and oxidative stress indicator in a porcine model. (**a**) Total nitric oxide (NO) concentration, (**b**) total antioxidant capacity (T-AOC), (**c**) superoxide dismutase (SOD) activity, (**d**) catalase activity, (**e**) glutathione-S-transferase (GSH) concentration, (**f**) malonyldialdehyde (MDA) concentration. Data in scatter plot are shown as mean ± SD. *t*-test was performed to find out significant difference between the two groups. **** denotes *p* ≤ 0.0001.

**Figure 6 animals-10-02312-f006:**
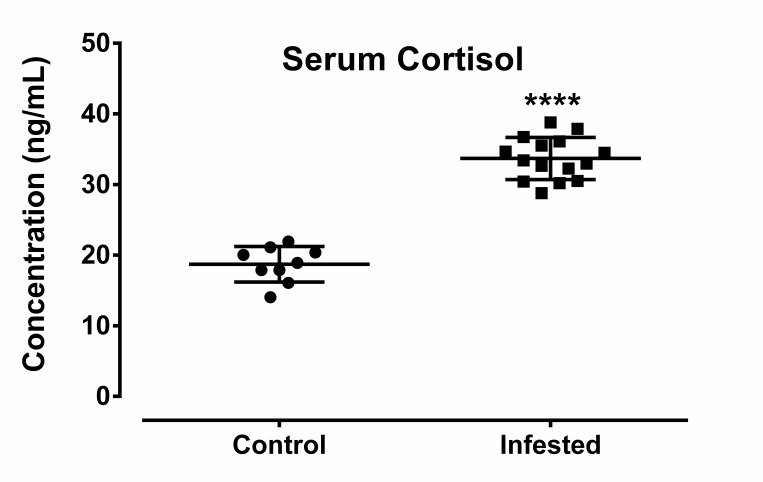
Effect of crusted scabieson serum cortisol concentration in a porcine model. Data in scatter plot are shown as mean ± SD. *t*-test was performed to find out significant difference between the two groups. **** denotes *p* ≤ 0.0001.

**Figure 7 animals-10-02312-f007:**
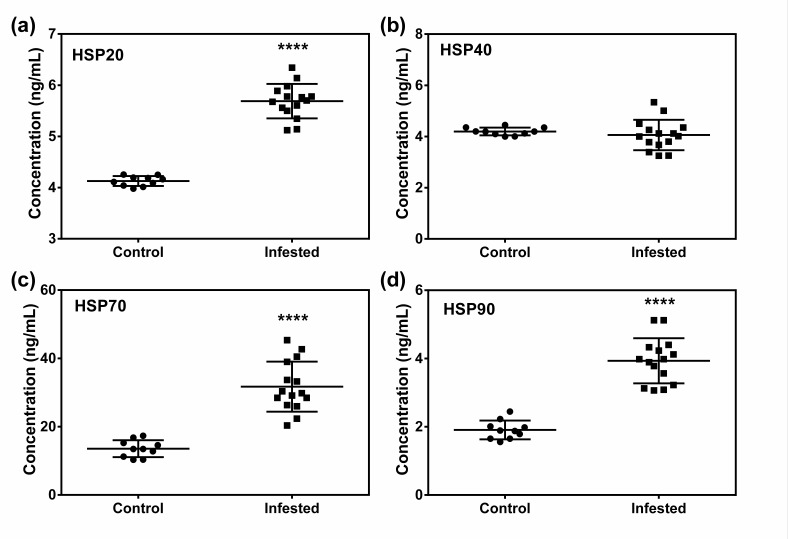
Effect of crusted scabies on serum heat shock proteins (HSPs) concentration in a porcine model. (**a**) HSP20, (**b**) HSP40, (**c**) HSP70, and (**d**) HSP90.Data in scatter plot are shown as mean ± SD. *t*-test was performed to find out significant difference between the two groups. **** denotes *p* ≤ 0.0001.

**Figure 8 animals-10-02312-f008:**
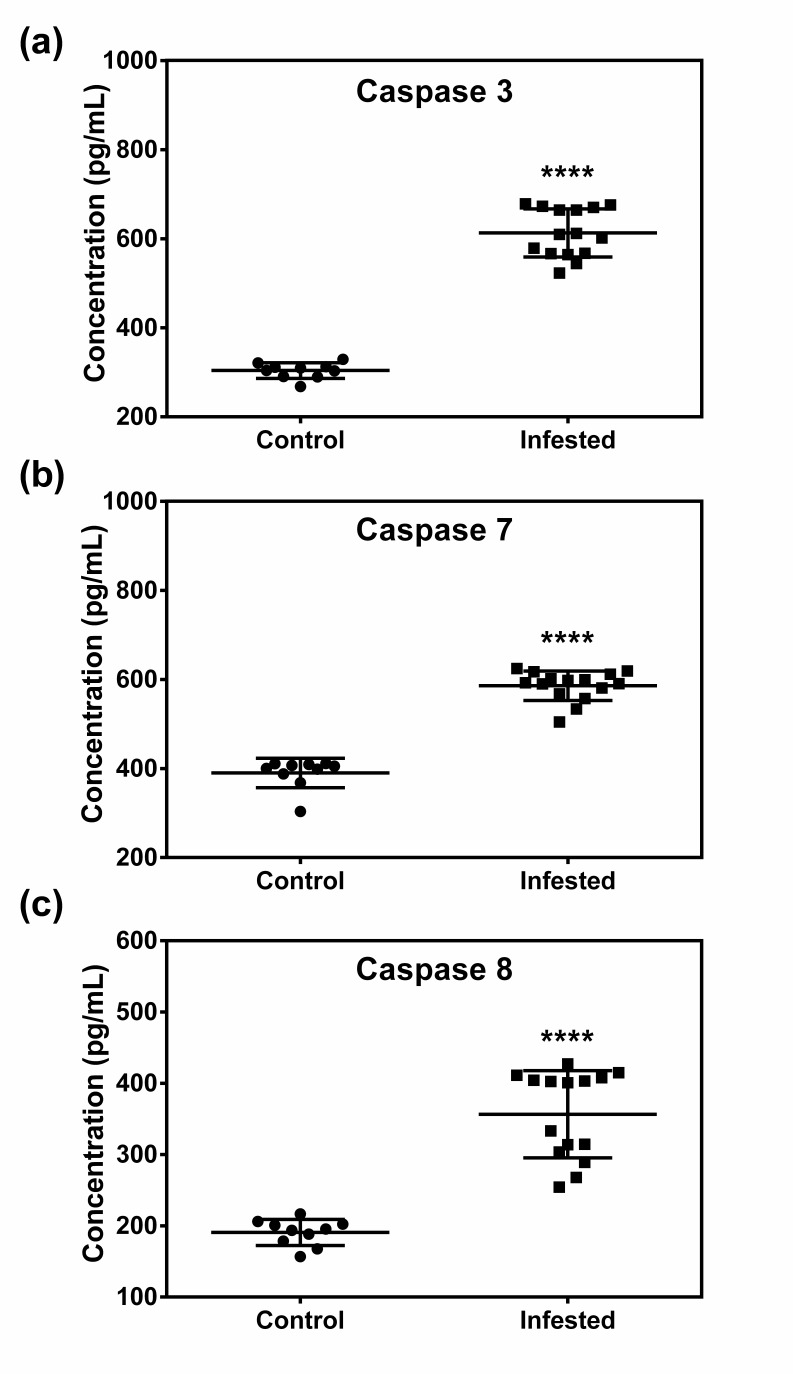
Effect of crusted scabieson serum levels of apoptotic markers in a porcine model. (**a**) Caspase 3, (**b**) caspase 7, (**c**) caspase 8. Data in scatter plot are shown as mean ± SD. *t*-test was performed to find out significant difference between the two groups. **** denotes *p* ≤ 0.0001.

**Figure 9 animals-10-02312-f009:**
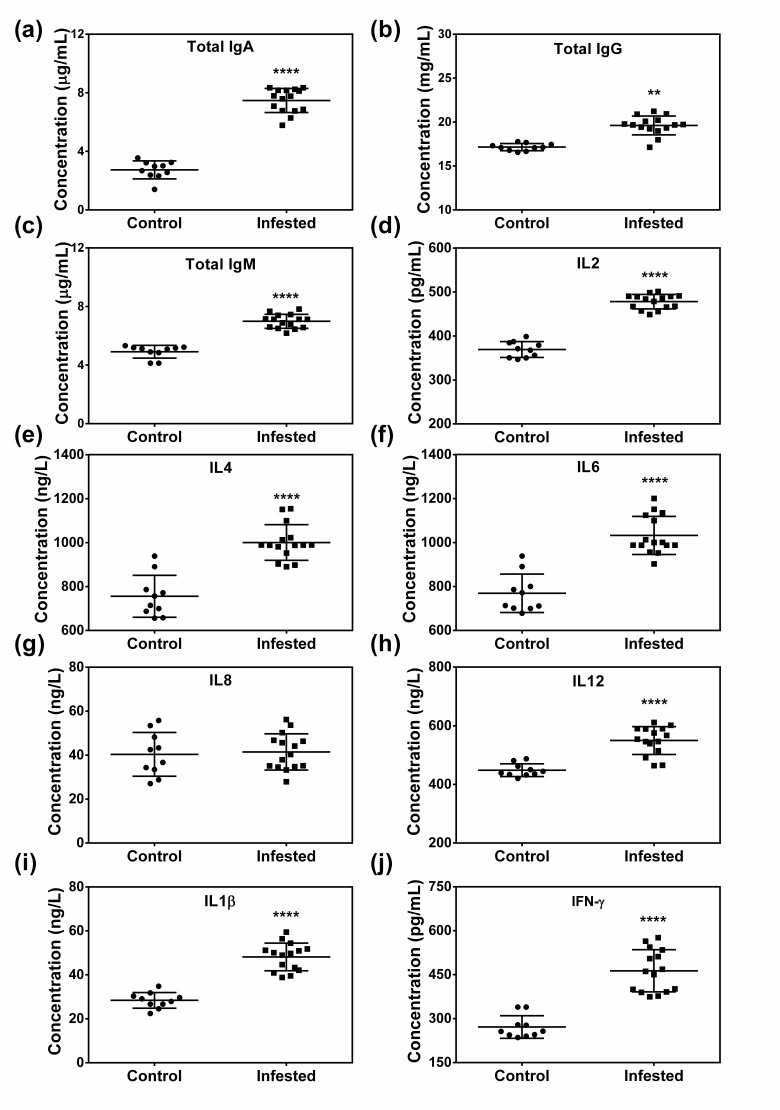
Effect of crusted scabieson total serum immunoglobulins and cytokine response in a porcine model. (**a**) Total IgA, (**b**) total IgG, (**c**) total IgM, (**d**) IL-2, (**e**) IL-4, (**f**) IL-6, (**g**) IL-8, (**h**) IL-12, (**i**) IL-1β, and (**j**) IFN-γ. Data in scatter plot are shown as mean ± SD. *t*-test was performed to find out significant difference between the two groups. ** denotes *p* ≤ 0.01, **** denotes *p* ≤ 0.0001.

**Figure 10 animals-10-02312-f010:**
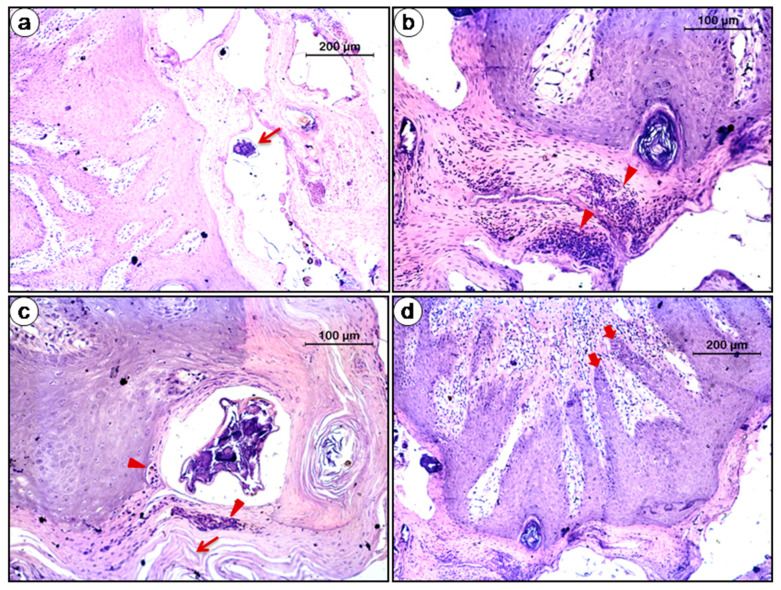
Histopathology of infested skin (**a**) Mite (arrow) with its exoskeleton and remnants prevailed in the tunnel of stratum cornium of epidermis (original magnification 100×, scale bar: 200 µm), (**b**,**c**) Parakeratotic hyperkeratosis characterized by upsurge in the thickness of the stratum corneum and excessive production of squames (arrow) followed by infiltration (arrow head) into epidermis suggested crust formation. There was infiltration of cells especially eosinophils (arrow head) into epidermis and dermal vascular area causing dermatitis (original magnification 200×, scale bar: 100 µm), (**d**) Psoriasiform hyperplasia: Note the epidermal projections (block arrow) into the dermis for interdigitaion with dermal papillae (original magnification 100×, scale bar: 200 µm).

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
