# Peer review of "Host-Parasite Interaction in Sarcoptes scabiei Infestation in Porcine Model with a Preliminary Note on Its Genetic Lineage from India"

_animals, 2020, doi:10.3390/ani10122312_

Round 1

Reviewer 1 Report

This manuscript by Arun Kumar De et al., describes several interesting points about the physiology of S. scabiei infection, including genetic aspects of this pathogen. In general the manscript is well written, easy to follow and brings new knowledge to this disease.

Major points:

  1. Why the personnel is not wearing gloves to mannipulate the infected animals (Fig 1A)?, it is a bad message to the readers.
  2. Figure 10 needs a detailed description on the figure legend, what are pointing the arrows and asterisks? Which magnification are the pictures?
  3. Do the authors have an idea around the time of infections of the pigs? Are they chronically infected?
  4. Do the authors think that the mixed Th1/Th2 response is associated with protection? Or may be a local immune response analysis is necessary to determine or improve the understanding of this disease? Please comment and include it in the discussion section.
  5. Eosinophils are indistinguishable in the histology sections.
  6. Regarding to the humoral response discussion, it is clear that as analyzed, it is not possbile to know whether or not the antibody response is S. scabiei specific. Please modify this point in your discussion section.
  7. The IgE discussion is not necessary, given that it was not measured in this work.
  8. Why do the authors consider the mixed Th1/Th2 response detected as a dysregulation of cytokine balance? What did they expect? No changes in the cytokine levels after a chronic infection? please explain.

Author Response

Reviewer 1:

Comments and Suggestions for Authors

This manuscript by Arun Kumar De et al., describes several interesting points about the physiology of S. scabiei infection, including genetic aspects of this pathogen. In general the manuscript is well written, easy to follow and brings new knowledge to this disease.

Response: We thank the reviewer for his/her enthusiasm in our study. We have addressed the points raised by the reviewer and included those in the revised manuscript. The authors would like to thank the reviewer for his/her comments for improvement of the manuscript.

Major points:

1. Why the personnel is not wearing gloves to mannipulate the infected animals (Fig 1A)?, it is a bad message to the readers.

Response: The authors agree with the reviewer. Accordingly, the figure (Fig 1A) has been modified.

2. Figure 10 needs a detailed description on the figure legend, what are pointing the arrows and asterisks? Which magnification are the pictures?

Response: Detailed description of figure legends has been included in the revised manuscript. Magnification has been indicated.

3. Do the authors have an idea around the time of infections of the pigs? Are they chronically infected?

Response: The animals were chronically infected.

4. Do the authors think that the mixed Th1/Th2 response is associated with protection? Or may be a local immune response analysis is necessary to determine or improve the understanding of this disease? Please comment and include it in the discussion section.

Response:Th1 immune response is generally associated with protection in most of the infectious diseases whereas Th2 immune response elicits high titers of antibody production and cell-mediated inflammation (Spellberg and Edwards, 2001). Moreover, allergic inflammatory diseases are dominated by Th2 immune response (Bhat et al., 2017). Induction of mixed Th1/Th2 immune response in the current study indicates that the immune response to mange infestation is complex and local immune response analysis is necessary for better understanding of the disease.

5. Eosinophils are indistinguishable in the histology sections.

Response: Though eosinophil in histopathology section could be easily recognised at 1000x magnification, still at 200x magnification it could be readable enough.

6. Regarding to the humoral response discussion, it is clear that as analyzed, it is not possbile to know whether or not the antibody response is S. scabiei specific. Please modify this point in your discussion section.

Response: The authors agree with the comment of the reviewer. According the text has been modified.

7. The IgE discussion is not necessary, given that it was not measured in this work.

Response: Modified as suggested.

8. Why do the authors consider the mixed Th1/Th2 response detected as a dysregulation of cytokine balance? What did they expect? No changes in the cytokine levels after a chronic infection? please explain.

Response: Cytokine milieu is the major player in polarization of native T cells into either Th1 or Th2 cells. IL-12 and IFN-γ are the key cytokines for polarization of Th0 to Th1 whereas IL-4 and IL-10 are crucial stimulatory cytokines for Th2 cell polarization respectively. In the present study, S. scabiei infestation was associated with up-regulation of IL-2, IFN-γ, IL-1β and IL-4 indicating both Th1 and Th2 response. Most probable cause of the mixed Th1/Th2 response might be dysregulation of cytokine balance due to modulation of immunity by S. scabiei released antigens or changes of cytokine levels after a chronic infection. This has been included in the discussion section.

Reviewer 2 Report

Line 91 – 95: The reference given in the text does not provide evidence for the stated fact. Secondly, my understanding is the zoonotic transmission of the clinical disease is not yet scientifically established. Therefore, I would suggest the authors to be cautious in the facts that they have stated in these line or back them up with strong literature.

Line 118 – Number of male and female animals in the control and the test groups should be mentioned as it plays a role differences on biochemical analyses.

Line 244 – All development stages should include male and female mites. So if the authors have seen and differentiate male and female mites, I suggest mentioning them in the text. If they haven’t, then the sentence should be modified only stating the life stages that have observed in the scraping.

Line 254 – As usually a figure should stand by itself apart from the body of the text, I suggest to include the microscopic image magnification which was used to take the images into the figure legend.

Line 269-270 – I suggest that the authors should discuss the Clade B in the figure 2a being closer to animal cluster than that of human cluster in the discussion.

Line 375 – The figure legend need to be formatted.

Line 376 – Figure 10a line arrow is not possibly directed to a mite considering the size according to the provided scale bar. Figure 10b line arrow looks more like a keratine pearl than a mite. These two need to be confirmed by a pathologist. Overall figure is overstained with haematoxylin to my taste.

General comments –

S. scabiei should be in italic font throughout the manuscript (special attention to the discussion)

The correct term for the animal S. scabiei infestation is “mange”. So I suggest to use the term where ever applicable. Scabies is the term used for S. scabiei infestation in human.

The data on feed intake and the feed conversion ratio would have been helpful to explain the dyslipidemia and serum biochemical parameters in depth. In brief, the drop of feed intake causes the increased serum creatinine which will also affect the renal and liver functions. It also affect the biochemical parameters related to nutrition being affected, apart from the direct effect of the infestation.

These biochemical and all the other parameters are evaluated only with the pigs who are infested with crusted scabies/mange. It should be highlighted throughout the manuscript. Generalising it to S. scabiei infestation is not accurate as the two forms (common and crusted) have significantly different pathology.

I suggest to provide reference values for the parameters measured as a supplemental table

Author Response

Reviewer 2

Comments and Suggestions for Authors

Line 91 – 95: The reference given in the text does not provide evidence for the stated fact. Secondly, my understanding is the zoonotic transmission of the clinical disease is not yet scientifically established. Therefore, I would suggest the authors to be cautious in the facts that they have stated in these line or back them up with strong literature.

Response: the portion has been rephrased.

Line 118 – Number of male and female animals in the control and the test groups should be mentioned as it plays a role differences on biochemical analyses.

Response: mentioned.

Line 244 – All development stages should include male and female mites. So if the authors have seen and differentiate male and female mites, I suggest mentioning them in the text. If they haven’t, then the sentence should be modified only stating the life stages that have observed in the scraping.

Response: Necessary modification has been made.

Line 254 – As usually a figure should stand by itself apart from the body of the text, I suggest to include the microscopic image magnification which was used to take the images into the figure legend.

Response: Microscopic image magnification has been provided.

Line 269-270 – I suggest that the authors should discuss the Clade B in the figure 2a being closer to animal cluster than that of human cluster in the discussion.

Response: Included in the discussion on the basis of low gene flow between the clades.

Line 375 – The figure legend need to be formatted.

Response: Formatted

Line 376 – Figure 10a line arrow is not possibly directed to a mite considering the size according to the provided scale bar. Figure 10b line arrow looks more like a keratine pearl than a mite. These two need to be confirmed by a pathologist. Overall figure is overstained with haematoxylin to my taste.

Response: Photographs were taken with a Leica DM500 microscope with leica application suite software aided scale bar. Arrow directed towards mite was well adjusted. A keratin pearl is a keratinized structure found in the regions where abnormal squamous cells form concentric layers. Keratin pearl formation in the invasive nests shows a growth pattern like cell clusters or irregular proliferating structures with central keratinization. In the present figure, the chitinous structure of mite was observed within the tunnel burrow in the stratum cornium of the epidermis as evident by the surrounding reaction from the prickle cell layer and lytic effect of keratine strand veil. This has been confirmed by a veterinary pathologist.

General comments –

S. cabieishould be in italic font throughout the manuscript (special attention to the discussion)

Response: Done

The correct term for the animal S. scabiei infestation is “mange”. So I suggest to use the term where ever applicable. Scabies is the term used for S. scabiei infestation in human.

Response: Done, wherever applicable

The data on feed intake and the feed conversion ratio would have been helpful to explain the dyslipidemia and serum biochemical parameters in depth. In brief, the drop of feed intake causes the increased serum creatinine which will also affect the renal and liver functions. It also affect the biochemical parameters related to nutrition being affected, apart from the direct effect of the infestation.

Response: The authors agree with the reviewer. The farm record (15 days data) on feed intake and FCR of infested and control animals didn't show any significant variation. Therefore, it is reasonable to assume that the dyslipidemia and changes in serum biochemical parameters  are due to infestation.

These biochemical and all the other parameters are evaluated only with the pigs who are infested with crusted scabies/mange. It should be highlighted throughout the manuscript. Generalising it to S. scabiei infestation is not accurate as the two forms (common and crusted) have significantly different pathology.

Response: Done

I suggest to provide reference values for the parameters measured as a supplemental table

Response: Provided (Table S2)
